# Dipeptidyl Peptidase 4 Stimulation Induces Adipogenesis-Related Gene Expression of Adipose Stromal Cells

**DOI:** 10.3390/ijms242216101

**Published:** 2023-11-08

**Authors:** Hsiao-Chi Lai, Pei-Hsuan Chen, Chia-Hua Tang, Lee-Wei Chen

**Affiliations:** 1Department of Surgery, Kaohsiung Veterans General Hospital, No. 386, Ta-Chung 1st Road, Kaohsiung 813, Taiwan; 2Institute of Emergency and Critical Care Medicine, National Yang Ming Chiao Tung University, No. 155, Sec. 2, Linong Street, Taipei 112, Taiwan; 3Department of Biological Sciences, National Sun Yat-Sen University, No. 70, Lien-Hai Road, Kaohsiung 804, Taiwan

**Keywords:** adipogenesis, PDGFRα, adipose, IL-10, FOXP3

## Abstract

Adipogenesis has emerged as a new therapeutic target for regulating metabolism and achieving anti-inflammatory and anti-atherosclerotic effects via the release of adiponectin. However, at present, the effects and mechanism of action of dipeptidyl peptidase 4 (DPP4) stimulation on adiponectin production and adipogenesis have not been clarified. Here, we investigated the effects of DPP4 stimulation with monocyte chemoattractant protein-1 (MCP-1) on platelet-derived growth factor receptor alpha (PDGFRα) expression in adipose tissue and blood adiponectin levels. Stromal vascular fractions (SVFs) purified from human subcutaneous adipose tissue and inguinal adipose tissue of obese and diabetic (*Lepr^db/db^*) mice were treated with 50 ng of MCP-1 and plasma from control (*Lepr^+/+^*) mice supplemented with 10 ng or 50 ng of MCP-1. Treatment of SVFs from human subcutaneous adipose tissues with 50 ng of MCP-1 significantly increased AdipoQ, DPP4, peroxisome proliferator-activated receptor gamma (PPARγ), fatty-acid-binding protein (FABP4), and SERBF1 mRNA expression. MCP-1-supplemented plasma increased adiponectin, CCAAT-Enhancer-binding protein alpha (C/EBPα), DPP4, IL-33, and PDGFRα mRNA expression and adiponectin and DPP4 protein expression, while decreasing the expression of IL-10 mRNA in SVFs compared with the levels in the plasma treatment group. MCP-1-supplemented plasma was shown to increase PPARγ, PPARγ2, adiponectin, DPP4, and FABP4 and decrease IL-10 mRNA expression in PDGFRα cells from adipose tissue. Meanwhile, MCP-1-supplemented plasma increased MCP-1, PDGFRα, TNFα, adiponectin, and IL-1β and decreased IL-10 and FOXP3 mRNA expression in DPP4 cells. Moreover, the injection of MCP-1-supplemented plasma into adipose tissue increased the proportion of DPP4^+^ cells among PDGFRα^+^ cells from adipose tissue and plasma adiponectin levels of *Lepr^db/db^* mice compared with the levels in the plasma injection group. Our results demonstrate that DPP4^+^ cells are important adipose progenitor cells. Stimulation of DPP4 with MCP-1 increases adipogenesis-related gene expression and the population of DPP4^+^ cells among PDGFRα^+^ cells in SVFs and blood adiponectin levels. DPP4 stimulation could be a novel therapy to increase local adipogenesis and systemic adiponectin levels.

## 1. Introduction

Dipeptidyl peptidase 4 (DPP4) cleaves the penultimate L-proline or L-alanine located in the N-terminal region of several polypeptides, such as glucagon-like peptide 1 (GLP-1), glucose-dependent insulinotropic polypeptide, neuropeptides, and chemokines [1]. DPP4 induces a direct proinflammatory effect on various cell types including macrophages, lymphocytes, and smooth muscle cells [2,3]. DPP4-positive interstitial progenitor cells give rise to intercellular adhesion molecule (ICAM1^+^) preadipocytes in both subcutaneous and visceral white adipose tissue (WAT) and contribute to basal and high-fat-diet-induced adipogenesis [4]. MCP-1, known as chemokine (CC-motif) ligand 2 (CCL2), is a member of the CC chemokine family and MCP-1-induced protein (MCPIP) is known to induce adipogenesis that causes an increase in the number of adipocytes [5]. A recent study indicated that the inflammation of adipose was necessary for the expansion and remodeling of healthy adipose tissue in mice [6]. However, studies have yet to reveal the relationship between MCP-1 and DPP4 in terms of the mechanisms of adipogenesis.

Adipocytes have been demonstrated to be derived from resident adipocyte progenitors in WAT [7]. Platelet-derived growth factor receptor alpha (PDGFRα) is a marker of adipocyte progenitors that can produce new functional adipocytes in vivo [8]. PDGFRα is expressed by mesenchymal cell populations and is involved in the development of diverse tissues [9]. Adipocyte stem cells have been defined by the expression of PDGFRα [10]. Recently, adipose stem and progenitor cells and PDGFRα^+^ were shown to produce IL-33 in WAT, whereas mesothelial cells served as an additional source of IL-33 in visceral WAT [11]. However, the molecular mechanisms controlling adipogenesis in a region-specific manner remain unclear.

Platelet-rich plasma (PRP) contains various growth factors and its therapeutic application has achieved positive outcomes for alopecia, skin rejuvenation, pigmentary disorders, lichen sclerosus, leprosy-induced peripheral neuropathy, scar revision, and plaque psoriasis [12]. It has been shown that PRP could increase the survival rate of fat cells and stem cell differentiation [13]. However, several studies demonstrated that PRP treatment was not associated with improved outcomes compared with the control treatment [14,15]. Adiponectin is a circulating hormone secreted from adipose tissue that exerts protective effects against metabolic syndrome [16], inflammation, and atherosclerosis [17,18]. Unlike adipokines such as leptin, plasma adiponectin levels are negatively associated with adiposity and decreased in type 2 DM, obesity, and insulin resistance [19,20]. Healthy WAT remodeling contributes to the maintenance of adiponectin levels in obesity [21]. Adiponectin treatment promotes adipocyte differentiation and reverses high-fat-diet-induced insulin resistance by increasing insulin-stimulated glucose uptake in muscle and adipose tissue [22]. However, adiponectin agonists have not been well established and their use in humans is still limited. Therefore, we hypothesized that the use of MCP-1 to activate dipeptidyl peptidase 4 in adipose SVFs would increase blood adiponectin levels. Thus, we examined whether the injection of MCP-1-supplemented plasma increased PDGFRα cells and adiponectin levels by stimulating the expression of DPP4 in adipose tissue. Stimulation of DPP4 with MCP-1 may become a novel therapeutic approach to induce PDGFRα in adipose tissue and systemic adiponectin levels.

## 2. Results

### 2.1. MCP-1 Treatment Increases Adipogenesis-Related Gene Expression of Adipose SVFs in Humans

To determine the effects of MCP-1 treatment on the mRNA expression of adipogenesis-related genes of SVFs from adipose tissue in humans, SVFs were harvested from subcutaneous adipose tissue of humans, purified, and treated with 10 or 50 ng of MCP-1 in vitro. Treatment of human SVFs with 50 ng of MCP-1 resulted in significant increases in the mRNA expression of AdipoQ, DPP4, PPARγ, and FABP4 of SVFs but did not change that of IL-10, C/EBPβ, and C/EBPα compared with the levels of those treated with PBS. Treatment of human SVFs with 50 ng of MCP-1 resulted in a significant increase in the mRNA expression of SERBF compared with the level of those treated with 10 ng of MCP-1. Taken together, these results indicate that treatment with 50 ng of MCP-1 increases the expression of AdipoQ, DPP4, PPARγ, and FABP4 mRNA in SVFs from human adipose tissue (Figure 1). It is thus suggested that MCP-1 treatment increases the expression of adipogenesis-related genes in human adipose SVFs.

### 2.2. MCP-1-Supplemented Plasma Enhances the Expression of Adipogenesis-Related Genes in SVFs of Adipose Tissue of Lepr^db/db^ Mice

To examine the mechanisms behind the effects of MCP-1 treatment on the expression of adipogenesis-related genes in SVFs from adipose tissue, we studied the mRNA expression of adipogenesis-related genes in SVFs harvested from inguinal adipose tissue from obese and diabetic (*Lepr^db/db^*) mice. SVFs were harvested from inguinal white adipose tissue, purified, and analyzed for the expression of IL-10, C/EBPα, adiponectin, IL-33, DPP4, and PDGFRα mRNA. Treatment of *Lepr^db/db^* SVFs with 10 ng MCP-1-supplemented *Lepr^+/+^* plasma induced significant increases in IL-33, adiponectin, and DPP4 mRNA expression and a reduction in IL-10 mRNA expression compared with the levels of those treated with *Lepr^+/+^* plasma (Figure 2). Treatment of *Lepr^db/db^* SVFs with 50 ng MCP-1-supplemented *Lepr^+/+^* plasma induced significant increases in DPP4, C/EBPα, adiponectin, and IL-33 mRNA expression and a decrease in IL-10 mRNA expression compared with the levels of those treated with *Lepr^+/+^* plasma (Figure 2). Moreover, SVFs from the adipose tissue of *Lepr^db/db^* mice treated with 50 ng MCP-1-supplemented *Lepr^+/+^* plasma demonstrated further increases in the expression of PDGFRα and IL-33 mRNA compared with those treated with 10 ng MCP-1-supplemented *Lepr^+/+^* plasma. Altogether, these results suggest that MCP-1-supplemented plasma increases PDGFRα, C/EBPα, adiponectin, DPP4, and IL-33 expression and decreases IL-10 mRNA expression of SVFs.

### 2.3. MCP-1-Supplemented Control Plasma Increases the Protein Expression of Adiponectin and DPP4 in SVFs from Adipose Tissue of Lepr^db/db^ Mice

To further determine the effects of MCP-1-supplemented plasma on the protein expression of pNF-κB, pJNK, adiponectin, and DPP4 in SVFs from adipose tissue, we examined the expression of pNF-κB, NF-κB, pJNK, JNK, adiponectin, and DPP4 proteins in SVFs derived from inguinal adipose tissue from *Lepr^db/db^* mice. SVFs were harvested from adipose tissue, treated with plasma supplemented with 10 or 50 ng MCP-1 or without supplementation, and then examined for protein expression of pNF-κB, NF-κB, pJNK, JNK, adiponectin, and DPP4. Treatment of SVFs from *Lepr^db/db^* mice with *Lepr^+/+^* plasma induced significant reductions in pNF-κB and pJNK protein expression compared with the levels of those treated with PBS (Figure 3A and Appendix A). Treatment of *Lepr^db/db^* SVFs with 50 ng MCP-1-supplemented *Lepr^+/+^* plasma induced significant increases in DPP4 and adiponectin protein expression compared with the levels of those treated with *Lepr^+/+^* plasma (Figure 3B). Taken together, these results suggest that MCP-1-supplemented plasma increases the DPP4 and adiponectin protein expression of SVFs from adipose tissue.

### 2.4. MCP-1-Supplemented Plasma Increases PPARγ, PPARγ2, Adiponectin, DPP4, and FABP4 mRNA Expression and Decreases IL-10 mRNA Expression in PDGFRα Cells from Adipose Tissue

To determine whether MCP-1-supplemented plasma increases the expression of adipogenesis-related genes and decreases IL-10 mRNA expression in PDGFRα cells, SVFs were harvested from adipose tissue of *Lepr^db/db^* mice and treated with 10 or 50 ng MCP-1-supplemented *Lepr^+/+^* plasma in vitro. PDGFRα cells were purified with microbeads and analyzed for IL-33, PPARγ, PPARγ2, IL-10, adiponectin, DPP4, and FABP4 mRNA expression. Treatment of SVFs from *Lepr^db/db^* mice with *Lepr^+/+^* plasma significantly increased the expression of IL-10 mRNA in PDGFRα cells compared with the level of those treated with PBS (Figure 4). Treatment of SVFs with 10 ng MCP-1-supplemented plasma resulted in the upregulation of adiponectin, PPARγ2, and DDP4 mRNA expression in PDGFRα cells and IL-33 mRNA expression in non-PDGFRα cells compared with the levels of those treated with plasma (Figure 4). Treatment of SVFs with 10 ng MCP-1-supplemented plasma resulted in the downregulation of IL-10 mRNA expression in PDGFRα cells compared with the level in those treated with plasma (Figure 4). Furthermore, treatment of SVFs with 50 ng MCP-1-supplemented *Lepr^+/+^* plasma resulted in increases in adiponectin, PPARγ, PPARγ2, and DPP4 mRNA expression in PDGFRα cells and IL-33 mRNA expression in non-PDGFRα cells compared with the levels in those treated with plasma. In addition, treatment of SVFs with 50 ng MCP-1-supplemented plasma resulted in the downregulation of IL-10 mRNA expression in PDGFRα cells compared with the level in those treated with plasma (Figure 4). Taken together, these results indicate that MCP-1-supplemented plasma increases adiponectin, PPARγ, PPARγ2, and DPP4 mRNA expression in PDGFRα cells and IL-33 mRNA expression in non-PDGFRα cells. MCP-1-supplemented plasma decreases the expression of IL-10 in PDGFRα cells. 

### 2.5. MCP-1-Supplemented Plasma Increases MCP-1, PDGFRα, TNFα, Adiponectin, and IL-1β mRNA Expression and Decreases IL-10 and FOXP3 mRNA Expression in DPP4 Cells from Adipose Tissue

To determine whether MCP-1-supplemented plasma increases PPARγ mRNA expression in DPP4 cells, SVFs were harvested from adipose tissue of *Lepr^db/db^* mice and treated with 10 or 50 ng MCP-1-supplemented *Lepr^+/+^* plasma in vitro. DPP4 cells were purified with microbeads and analyzed for MCP-1, IL-33, PDGFRα, TNFα, IL-10, adiponectin, Foxp3, and IL-1β mRNA expression. Treatment of SVFs from *Lepr^db/db^* mice with *Lepr^+/+^* plasma significantly decreased PDGFRα and increased IL-10 mRNA in DPP4 cells compared with the levels in those treated with PBS (Figure 5). Treatment of SVFs with 10 ng MCP-1-supplemented plasma resulted in the upregulation of PDGFRα and downregulation of IL-10 and Foxp3 mRNA expression in DPP4 cells compared with the levels in those treated with plasma (Figure 5). Treatment of SVFs with 50 ng MCP-1-supplemented plasma resulted in the upregulation of MCP-1, PDGFRα, TNFα, adiponectin, and IL-1β and downregulation of IL-10 and Foxp3 mRNA expression in DPP4 cells compared with the levels in those treated with plasma (Figure 5). Furthermore, 50 ng MCP-1-supplemented *Lepr^+/+^* plasma resulted in further increases in MCP-1, PDGFRα, and adiponectin mRNA expression in DPP4 cells compared with the levels in those treated with 10 ng MCP-1-supplemented *Lepr^+/+^* plasma. Taken together, these results indicate that MCP-1-supplemented plasma increases PDGFRα and adiponectin and decreases IL-10 mRNA in DPP4 cells. The results also show that MCP-1-supplemented plasma increases the expression of MCP-1, PDGFRα, and adiponectin mRNA in DPP4 cells in a dose-dependent manner.

### 2.6. Injection of MCP-1-Supplemented Plasma into Adipose Tissue of Lepr^db/db^ Mice Increases DPP4^+^ Cells among PDGFα^+^ Cells

To examine which cells of SVFs demonstrated increased PDGFRα expression after the injection of MCP-1-supplemented plasma in *Lepr^db/db^* mice, we conducted flow cytometry analysis to assess the numbers of CD8 and CD11b cells in the SVFs from adipose tissue. We found that such injection significantly increased the number of CD11b cells in isolated PDGFRα^+^ cells compared with the number of CD8 cells (Figure 6A,B). There were no significant differences in CD11b cells and CD8 cells between the treatment groups. This indicates that most PDGFRα^+^ cells are macrophages. We further investigated whether DPP4 cells were included among isolated PDGFRα^+^ cells after the injection of MCP-1-supplemented plasma into *Lepr^db/db^* mice. For this purpose, we analyzed the numbers of DPP4^+^ cells in the SVFs from adipose tissue by flow cytometry. Interestingly, we found that DPP4^+^ cells among isolated PDGFRα^+^ cells were significantly increased in *Lepr^db/db^* mice compared with those in *Lepr^+/+^* mice. The injection of *Lepr^+/+^* plasma into the adipose tissue of *Lepr^db/db^* mice significantly decreased DPP4^+^ cells among PDGFα^+^ cells compared with that in the PBS group. The injection of 10 ng or 50 ng MCP-1-supplemented *Lepr^+/+^* plasma into the adipose tissue of *Lepr^db/db^* mice significantly increased DPP4^+^ cells among PDGFRα^+^ cells compared with the level in the control plasma injection group (Figure 6C,D). Taken together, these results suggest that control plasma decreased and MCP-1-supplemented plasma increased the numbers of DPP4^+^ cells among PDGFα^+^ cells from adipose tissue and that most PDGFRα^+^ cells are macrophages. 

### 2.7. Injection of MCP-1-Supplemented Plasma into Adipose Tissue of Lepr^db/db^ Mice Increases PDGFRα mRNA Expression in Adipose Tissue Macrophages 

To determine the effects of MCP-1-supplemented plasma on PDGFRα mRNA expression in macrophages in adipose tissue, plasma supplemented with 0, 10, or 50 ng of MCP-1 was injected into the adipose tissue of *Lepr^db/db^* mice. ATMs were purified with microbeads and analyzed for the expression of PDGFRα mRNA. The injection of *Lepr^+/+^* plasma into the adipose tissue of *Lepr^db/db^* mice significantly decreased the expression of PDGFRα mRNA in adipose tissue macrophages compared with that in the PBS injection group (Figure 7A). The injection of 10 ng or 50 ng MCP-1-supplemented plasma into the adipose tissue of *Lepr^db/db^* mice significantly increased the expression of PDGFRα mRNA in adipose tissue macrophages compared with that in the plasma injection group. The injection of 0, 10, or 50 ng MCP-1-supplemented plasma did not change the expression of PDGFRα mRNA in non-adipose tissue macrophages compared with that in the plasma injection group. Taken together, these results indicate that MCP-1-supplemented plasma increases the expression of PDGFRα mRNA in ATMs and that PDGFRα mRNA is mostly expressed in adipose tissue macrophages of SVFs.

### 2.8. Injection of MCP-1-Supplemented Plasma into Adipose Tissue of Lepr^db/db^ Mice Enhances Plasma Adiponectin Levels

To examine the effects of MCP-1-supplemented plasma on blood adiponectin levels, plasma with 0, 10, or 50 ng MCP-1 supplementation was injected into the adipose tissue of *Lepr^db/db^* mice. Blood was harvested after 1 week and plasma adiponectin levels were assayed. *Lepr^db/db^* mice demonstrated a significant increase in blood adiponectin levels compared with *Lepr^+/+^* mice. The injection of *Lepr^+/+^* plasma into the adipose tissue of *Lepr^db/db^* mice significantly reduced plasma adiponectin levels compared with that in the PBS injection group (Figure 7B). The injection of 10 ng MCP-1-supplemented plasma into the adipose tissue of *Lepr^db/db^* mice mildly increased the plasma adiponectin levels compared with that in the plasma injection group. The injection of 50 ng MCP-1-supplemented plasma to the adipose tissue of *Lepr^db/db^* mice significantly induced plasma adiponectin levels compared with that in the plasma injection group. These results suggest that the injection of control plasma into adipose tissue of obese mice reduces blood adiponectin levels, while the injection of 50 ng MCP-1-supplemented plasma increases plasma adiponectin levels.

## 3. Discussion

Adipose tissue expands through either the differentiation of new adipocytes (adipogenesis) or the hypertrophy of adipocytes from adipocyte precursor cells (APCs). Adipogenesis has emerged as a new therapeutic target for its beneficial effects in regulating metabolism and exerting anti-inflammatory and anti-atherosclerotic activities [16]. Adiponectin produced by adipocytes is a biologically active polypeptide that exerts its beneficial physiological effects in the regulation of metabolism [16]. Macrophages in the visceral adipose tissue demonstrated increased proliferation in obesity. MCP-1 treatment induced macrophage cell division in adipose tissue, whereas MCP-1 deficiency in vivo reduced adipose tissue proliferation [23]. DPP4-positive interstitial progenitor cells contribute to basal and high-fat-diet-induced adipogenesis [4]. Therefore, we examined whether MCP-1-supplemented control plasma induced the expression of adipogenesis-related genes in SVFs and adiponectin levels through stimulating DPP4. First, MCP-1 treatment increases the expression of AdipoQ, DPP4, PPARγ, and FABP4 mRNA in SVFs from human adipose tissue in vitro. This suggests that MCP-1 could be used to increase adipogenesis-related gene expression of adipose SVFs in humans. Second, MCP-1-supplemented plasma increased adiponectin, C/EBPα, DPP4, IL-33, and PDGFα mRNA expression and adiponectin and DPP4 protein expression in the SVFs of adipose tissue in vitro. This suggests that treatment with such plasma increases adipogenesis-related gene expression and DPP4 as well as adiponectin protein expression in SVFs. Third, MCP-1-supplemented plasma increased PPARγ, PPARγ2, adiponectin, DPP4, and FABP4 and decreased IL-10 mRNA expression in PDGFRα cells from adipose tissue. These results suggest that MCP-1-supplemented plasma increases the expression of adipogenesis-related genes in PDGFRα cells and IL-33 mRNA expression in non-PDGFRα cells. Fourth, MCP-1-supplemented plasma increased MCP-1, PDGFα, TNFα, adiponectin, and IL-1β mRNA expression and decreased IL-10 and FOXP3 mRNA expression in DPP4 cells from adipose tissue. This suggests that MCP-1-supplemented plasma increases adipogenesis-related gene expression and decreases IL-10 mRNA in DPP4 cells. Moreover, the injection of control plasma decreased PDGFRα mRNA expression and the injection of MCP-1-supplemented plasma into adipose tissue of *Lepr^db/db^* mice increased PDGFRα mRNA expression in adipose tissue macrophages. The injection of control plasma decreased plasma adiponectin levels, while the injection of MCP-1-supplemented plasma into adipose tissue of *Lepr^db/db^* mice increased these levels. 

Radiation-induced soft tissue defects have become a common problem in cancer treatment [24]. However, there were only limited methods could be used to increase local adipogenesis or fat grafting survival. The PPARγ agonist thiazolidinedione (TZD) is used as a proadipogenic compound, but its use remains controversial because it is associated with certain cardiac side effects and weight gain [3]. Defects in adipocyte differentiation induce pathologic adipose tissue change, fibrosis, immune cell activity, and metabolic syndrome. Adipogenesis supports healthy adipose tissue remodeling in obesity. Stimulating adipogenesis during caloric excess can drive healthy WAT remodeling. Therefore, healthy WAT remodeling contributes to the maintenance of adiponectin levels in obesity [21]. Adiponectin treatment reverses high-fat-diet-induced insulin resistance through the increase in insulin-stimulated glucose in adipose tissue and muscles in mice [22]; however, adiponectin agonists have not been well characterized and their availability for therapeutic purposes in humans is still limited. Taken together, our findings suggest that MCP-1-supplemented plasma stimulates adipogenesis-related gene expression and adiponectin levels through increases in DPP4 and PDGFRα. MCP-1-supplemented plasma may be used as a new therapeutic strategy to increase local adipogenesis in treating lipodystrophy and soft tissue defects. 

A targeted reduction in DPP4 expression in the liver with siRNAs reduced hepatic mRNA expression of PPARγ. This suggests that a targeted reduction in DPP4 expression in the liver may improve hepatic lipid metabolism [25]. Mice lacking the DPP4 gene were difficult to develop obesity after a high-fat diet [26]. Ablation of the DPP4 gene was also associated with reduced lipogenesis [26]. DPP4 inhibitors are mainly weight-neutral in patients with type 2 diabetes in combination and monotherapy clinical trials [27]. This effect may contribute to the inhibition of intestinal fat extraction [28]. Our data demonstrate that DPP4 stimulation with MCP-1 increases adipogenesis-related gene expression and the population of DPP4^+^ cells among PDGFRα^+^ cells in SVFs and blood adiponectin levels. This suggests that DPP4 stimulation may induce local adipogenesis through the increase in adipogenesis gene expression without weight gain. 

PDGFRα is a marker of adipocyte progenitors that can produce new functional adipocytes in vivo [8]. SCA1^+^PDGFRα^+^ mesenchymal cells (MSCs) are known as adipogenic progenitor cells and have been assigned a variety of different functions [29]. However, the dynamics of adipocyte progenitors during adipose expansion have not been well characterized. Here, we characterized PDGFRα cells in SVFs of adipose tissue during adipogenesis. First, our data demonstrated that MCP-1-supplemented plasma increased the expression of PPARγ, PPARγ2, adiponectin, DPP4, and FABP4 and decreased IL-10 mRNA expression in PDGFRα cells from adipose tissue. Moreover, MCP-1-supplemented plasma increased the expression of IL-33 mRNA in non-PDGFRα cells. This suggests that the increase in adipogenesis-related gene expression in PDGFRα^+^ cells of adipose tissue will also increase the expression of IL-33 in non-PDGFRα cells and decrease the expression of IL-10. Second, we further characterized the PDGFRα cells by flow cytometry and found that most PDGFRα^+^ cells are CD11b cells. This suggests that most PDGFRα^+^ cells are macrophages. Mounting evidence suggests a role for yolk-sac-derived macrophages in adipocyte formation and expansion in adult mice [30]. Our data further demonstrated that the injection of MCP-1-supplemented plasma into adipose tissue of *Lepr^db/db^* mice increased PDGFRα mRNA expression in adipose tissue macrophages. This suggests that MCP-1-supplemented plasma increases the expression of PDGFRα mRNA in SVFs and that PDGFRα+ cells are mainly adipose tissue macrophages. Taken together, our data suggest that PDGFRα cells are important adipose progenitor cells and that stimulating DPP4 induces adipogenesis-related gene expression in PDGFRα cells. Increasing the expression of genes related to adipogenesis also decreases IL-10 expression in PDGFRα cells and increases IL-33 in non-PDGFRα cells.

Recently, a highly proliferative dipeptidyl peptidase-4 (DPP4^+^) APC population was identified, which gives rise to highly adipogenic ICAM1^+^ preadipocytes as well as a CD142^+^ adipogenic cell population [4]. To further characterize DPP4 cells and the effects of DPP4 stimulation on adipogenesis, MCP-1-supplemented plasma was used to stimulate DPP4 of SVFs. First, MCP-1 treatment increased DPP4 mRNA expression in SVFs from human adipose tissue in vitro. Second, MCP-1-supplemented plasma increased DPP4 mRNA expression and adiponectin and DPP4 protein expression in the SVFs of adipose tissue in vitro. This suggests that treatment with MCP-1-supplemented plasma increases adipogenesis-related gene expression and DPP4 protein expression in SVFs. Third, MCP-1-supplemented plasma increased DPP4 mRNA expression in PDGFRα cells and adipogenesis-related gene expression in DPP4^+^ cells. Furthermore, MCP-1-supplemented plasma significantly increased DPP4^+^ cells among PDGFRα^+^ cells in SVFs. Taken together, our findings suggest that DPP4^+^ cells are important adipose progenitor cells and that stimulating DPP4 with MCP-1-supplemented plasma induces adipogenesis-related gene expression, PDGFRα+ cells, and blood adiponectin levels. Stimulating DPP4 in adipose tissue may become a novel therapeutic strategy to induce local adipogenesis to enhance fat graft survival in the reconstruction of soft tissue defects.

Adipose progenitor cells-derived IL-33 maintains innate and adaptive immune cell activity and immune homeostasis in WAT [11]. Recently, WAT-resident mesenchyme-derived stromal cells were found to be the dominant producers of IL-33 [10,11]. However, the cellular sources and role of IL-33 in regulating adipogenesis remain incompletely understood. Our data demonstrated that MCP-1 supplementation induced the upregulation of adiponectin, mRNA expression in PDGFRα cells, and IL-33 mRNA expression in non-PDGFRα cells. IL-10 is a type II cytokine with anti-inflammatory activity, the loss of which is associated with autoimmune diseases [31]. However, the role of IL-10 in adipogenesis has not been well defined. Our data suggest that increased adipogenesis is associated with the increased expression of adipogenesis-related genes in PDGFRα cells and IL-33 mRNA expression in non-PDGFRα cells, along with a decrease in IL-10 mRNA expression in PDGFRα cells. 

PRP has been shown to enhance graft survival rate [13]. However, PRP has been demonstrated to increase bone formation and restrain adipogenesis by downregulating PPARγ and FABP4 in rat models of steroid-associated necrosis of the femoral head [32]. Our study demonstrated that control plasma treatment significantly reduced PDGFRα expression in adipose tissue macrophages and plasma adiponectin levels. Moreover, control plasma treatment increased the expression of IL-10 mRNA in PDGFRα and DPP4 cells of SVFs from adipose tissue. This indicates that control plasma reduces adipogenesis and inflammation of adipose tissue through decreased PDGFRα and increased IL-10. MCP-1 plays an important role in the inflammatory process, where it increases the expression of other inflammatory factors and cells. This results in the progression of many disorders by increasing the infiltration and migration of inflammatory cells such as macrophages to the inflammation site [33]. Our results suggest that adding MCP-1 to PRP may provide a valuable therapeutic approach to increase fat graft survival and adipogenesis in lipodystrophy patients by increasing PDGFRα and decreasing IL-10 expression of adipose tissue.

Our study had several limitations. We did not compare the effects of MCP-1-supplemented plasma with a PPARγ agonist in terms of the adipogenesis-related gene expression of SVFs and systemic adiponectin levels. We did not use IL-33- and IL-10-deficient mice to clarify the roles of IL-33 and IL-10 in adipogenesis and PDGFRα upregulation. Further study is required to proceed beyond these limitations.

## 4. Materials and Methods

### 4.1. Animals

We purchased *Lepr^db/+^* mice on the C57BL/6J genetic background from the Jackson Laboratory (Bar Harbor, ME, USA). Heterozygous *Lepr^db/+^* mice were crossed to obtain lean and control *Lepr*^+/+^ mice and obese as well as diabetic *Lepr^db/db^* mice. Obese and diabetic (*Lepr^db/db^*) and lean and nondiabetic (*Lepr^+/+^*) mice were used to study the differences in mRNA expression of inflammatory and adipogenesis-related genes in SVFs from adipose tissue. *Lepr^db/db^* mice carry a mutation in the gene encoding the leptin receptor and become obese around three to four weeks of age, and elevated plasma blood sugar and insulin can be observed at four to seven weeks of age. All mice had access to water and food *ad libitum* and were fed a standard laboratory diet (1324 TPF; Atromin; Large Germany; 11.9 kJ/g, 19% crude protein, 4% crude fat, 6% crude fiber). All animal experimental procedures were reviewed and approved by the Institutional Animal Care and Use Committee (IACUC) of Kaohsiung Veterans General Hospital. 

### 4.2. Humans

Human adipose tissue was obtained from normal subcutaneous adipose tissue during an operation involving the excision of lipoma from six female patients (aged 20–40 years, BMI 19.0–25.0 kg/m^2^), after the provision of written informed consent. The protocol conformed to the guidelines of the 1975 Declaration of Helsinki and was approved by the ethics committee of Kaohsiung Veterans General Hospital (VGHKS19-CT2-20).

### 4.3. Preparation of Stromal Vascular Fractions 

SVFs were prepared from human, *Lepr^db/db^*, and *Lepr^+/+^* mice. Vascular adipose tissue was isolated from subcutaneous adipose tissue of humans and bilateral inguinal adipose tissue of *Lepr^db/db^* or *Lepr^+/+^* mice and minced into small pieces. These tissues were then digested with collagenase VIII (Cat# C2139, Sigma-Aldrich, St. Louis, MO, USA) in ice-cold HBSS (2 mg/mL) for 20 min at 37 °C. Adipose tissue from humans was digested with collagenase 2 (Sigma-Aldrich, Cat# C6885) and adipose tissue from mice was digested with collagenase 8 (Sigma-Aldrich, Cat# C2139), respectively, in ice-cold HBSS (2 mg/mL) for 20 min at 37 °C. After passing through a 100 μm cell strainer, cells were centrifuged at 1700 rpm for 10 min, and the cell pellets were retrieved as SVFs. 

### 4.4. In Vitro Treatment of SVFs

Ghorpade et al. found that 10% plasma from diet-induced obesity (DIO) mice induced a higher expression of IL-6 and MCP-1 than 10% plasma from lean mice in SVFs of DIO mice [3]. For in vitro treatment, 1 mL of PBS, and 10% plasma with or without supplementation of MCP-1 at 10 ng or 50 ng, was added to the equal amount of SVFs (2 × 10^7^ cells), and the mixture was then incubated for 3 h at 37 °C [3]. Samples were centrifuged at 1700 rpm for 10 min and washed with phosphate-buffered saline (PBS). The pellets were harvested after centrifugation and subjected to analysis. 

### 4.5. Isolation of PDGFRα Cells from SVFs with Microbeads

The SVFs were pelleted by centrifugation, washed once with 1 mL of buffer, and resuspended in 190 μL of PBS–fetal bovine serum. The cells were then incubated with 10 μL of streptavidin-conjugated magnetic microbeads (1:10) (130-101-502, Miltenyi Biotec, San Diego, CA, USA) at 4 °C for 15 min. The cells were rinsed once with PBS–bovine serum albumin (BSA), and PDGFRα cells were isolated using magnetic separation columns (Miltenyi Biotec, 130-042-201), resuspended in buffer, and analyzed by qPCR. Non-PDGFRα (NPDGFRα) cells were isolated from the washing solution.

### 4.6. Isolation of DPP4 Cells from SVFs with Microbeads

The SVFs were pelleted by centrifugation, washed once with 1 mL of buffer, resuspended in 190 μL of PBS–fetal bovine serum (FBS), and labeled with the primary mouse IgG1 antibody DPP4 (# GTX84602, GeneTex, Irvine, CA, USA) according to the manufacturer’s recommendations. The cells were washed with 1 mL of PBS–fetal bovine serum (FBS). Samples were then centrifuged at 1700 rpm for 10 min. The cell pellet was resuspended in 80 µL of buffer, incubated with 20 μL of Anti-Mouse IgG1 MicroBeads (Miltenyi Biotec, 130-047-101), and incubated for 15 min in the refrigerator (2–8 °C). The cells were rinsed once with PBS–BSA, and DPP4 cells were isolated using magnetic separation columns (Miltenyi Biotec, 130-042-201) and resuspended in a buffer and analyzed with qPCR. Non-DPP4 (NDPP4) cells were isolated from the washing solution.

### 4.7. Isolation of Adipose Tissue Macrophages (ATMs) from SVFs with Microbeads

The SVF pellets were washed with 1 mL of buffer and resuspended in 190 μL of PBS–fetal bovine serum (FBS). The cells were incubated with 10 μL streptavidin-conjugated magnetic microbeads (1:10) (Miltenyi Biotec, 130-110-443, San Diego, CA, USA) for 15 min at 4 °C. The cells were washed once with PBS–BSA, and ATMs were isolated using magnetic separation columns (Miltenyi Biotec, 130-042-201), resuspended in a buffer, and analyzed with qPCR. Non-ATM (NATM) were harvested from the washing solution.

### 4.8. RNA Isolation and Quantitative Real-Time Polymerase Chain Reaction (qPCR)

Total RNA was isolated from mouse samples using total RNA Miniprep Purification Kits (GeneMark, GMbiolab, Taichung, Taiwan) according to the manufacturer’s instructions. Total RNA was reverse-transcribed to cDNA using an RT kit (Invitrogen, Carlsbad, CA, USA, Lot# 2234812). Primer pairs were synthesized by Integrated DNA Technologies (Coralville, IA, USA). For qPCR assay, 200 ng of the cDNA template was added to 25 μL of mix containing 12.5 μL of 2× Fast SYBR Green Master Mix (Applied Biosystems, Cat# 4385612, Waltham, MA, USA), 2.5 μL of sense and anti-sense primers (25 μM), 2 μL of sample, and 8 μL of sterile water. The amplification was performed in a StepOnePlus™ Real-Time PCR System (Applied Biosystems 7300). 

### 4.9. Western Immunoblots

The expression of DPP4 (GeneTex # GTX84602, Irvine, CA, USA), JNK (Cell Signaling, # 9252, Danvers, MA, USA), pJNK (Cell Signaling, # 9251), NF-κB (Cell Signaling, # 8242), pNF-κB P65 (Cell Signaling, # 3033), and adiponectin (Abcam, # ab22554, Cambridge, UK) proteins in the SVFs was examined by Western immunoblot analysis. The harvested samples were weighed and homogenized in a protein extraction buffer (Sigma, St. Louis, MO, USA), containing a proteinase inhibitor cocktail (Roche, Basel, Switzerland). The homogenized samples were subjected to SDS-PAGE for 2 h at 50 to 100 V, and the protein was transferred to the nitrocellulose membrane. The membranes were blocked with 5% non-fat milk in TBST buffer (10 mM Tris-HCl, pH 7.5, 150 mM NaCl and 1.2% Tween 20) for 1 h and incubated with specific primary antibodies for 1 h at room temperature. The membranes were then washed with TBST buffer followed by incubation with the secondary antibodies in the blocking buffer. After washing with TBST buffer, the protein bands were detected by enhanced chemiluminescence (ECL) detection reagent (Millipore, Burlington, MA, USA).

### 4.10. Flow Cytometry Analysis

Cells in the SVFs were suspended in staining buffer (PBS containing 0.5% BSA and 2 mM ethylenediaminetetraacetic acid) and then incubated with CD11b (BioLegend, San Diego, CA, USA, clone: M1/70), PDGFRα (BioLegend, clone: APA5), and DPP4 (BioLegend, clone: H194-112) antibodies or the control isotypes at 4 °C. Thirty minutes later, cells were washed twice and resuspended in the staining buffer. The cells were analyzed using an Attune NxT Flow Cytometer (ThermoFisher, Waltham, MA, USA). The data analysis was performed using FlowJo (Tree Star, Ashland, OR, USA). In another experiment, cells in the SVFs were incubated with CD11b, PDGFRα, and CD8a (BioLegend, clone: 53-6.7) to identify CD11b and CD8a cells among the PDGFRα^+^ cells in SVFs.

### 4.11. In Vivo Injection of MCP-1-Supplemented Plasma into Adipose Tissue

*Lepr^+/+^* mice received 1 mL PBS injection into adipose tissue over the bilateral inguinal area. *Lepr^db/db^* mice were randomly divided into the following four groups: Group I received 1 mL PBS injection into adipose tissue over the bilateral inguinal area; Group II received an injection of 1 mL 10% plasma from *Lepr^+/+^* mice into adipose tissue over the bilateral inguinal area; Group III received an injection of 1 mL 10% plasma from *Lepr^+/+^* mice supplementation with 10 ng MCP-1 into adipose tissue over the bilateral inguinal area; and Group IV received an injection of 1 mL 10% plasma from *Lepr^+/+^* mice supplementation with 50 ng MCP-1 into adipose tissue over the bilateral inguinal area. Seven days after injection, the animals were euthanized and plasma was harvested for analysis.

### 4.12. Enzyme-Linked Immunosorbent Assay (ELISA)

Adiponectin was examined by using a mouse ELISA kit from Invitrogen (# KMP0041). The blood was centrifuged at 1000× *g*, 4 °C for 15 min and the serum was harvested for use. The ELISA plates were coated with 100 μL capture antibody at 4 °C overnight. After washing, 200 μL of assay dilution buffer was added for blocking for 1 h at room temperature. The samples and serial dilutions of standards were added and incubated for overnight at 4 °C. After incubating with the antibody, avidin-HRP was added and incubated for 30 min at room temperature. The substrate, 3,3′,5,5′-tetramethylbenzidine (TMB), was added and incubated for 15 min. About 100 µL of stop solution was added to stop the reaction, and the plate was subjected to measurement of absorbance at 450 nm with an ELISA reader. 

### 4.13. Statistical Analysis

Data were analyzed by an unpaired *t*-test for comparisons between two groups or by one-way analysis of variance (ANOVA) followed by Tukey’s multiple comparison test for comparisons between multiple groups. All data in the figures and texts were expressed as mean ± standard error of the mean, and *p* values less than 0.05 are considered statistically significant.

## 5. Conclusions

We demonstrated the effects and mechanism of action of dipeptidyl peptidase 4 (DPP4) stimulation on adiponectin production and PDGFRα cells in adipose tissue. MCP-1 increases the expression of adipogenesis-related genes in SVFs of humans. MCP-1-supplemented plasma increased the mRNA expression of adipogenesis-related genes, the protein expression of adiponectin and DPP4, and decreased the expression of IL-10 in SVFs. MCP-1-supplemented plasma increased the expression of adipogenesis-related genes and decreased IL-10 mRNA expression in adipose PDGFRα^+^ cells and DPP4^+^ cells. DPP4 cells are important adipose progenitor cells. Stimulating DPP4 cells with MCP-1-supplemented plasma induces adipogenesis-related gene expression of PDGFRα cells and blood adiponectin levels.

## Figures and Tables

**Figure 1 ijms-24-16101-f001:**
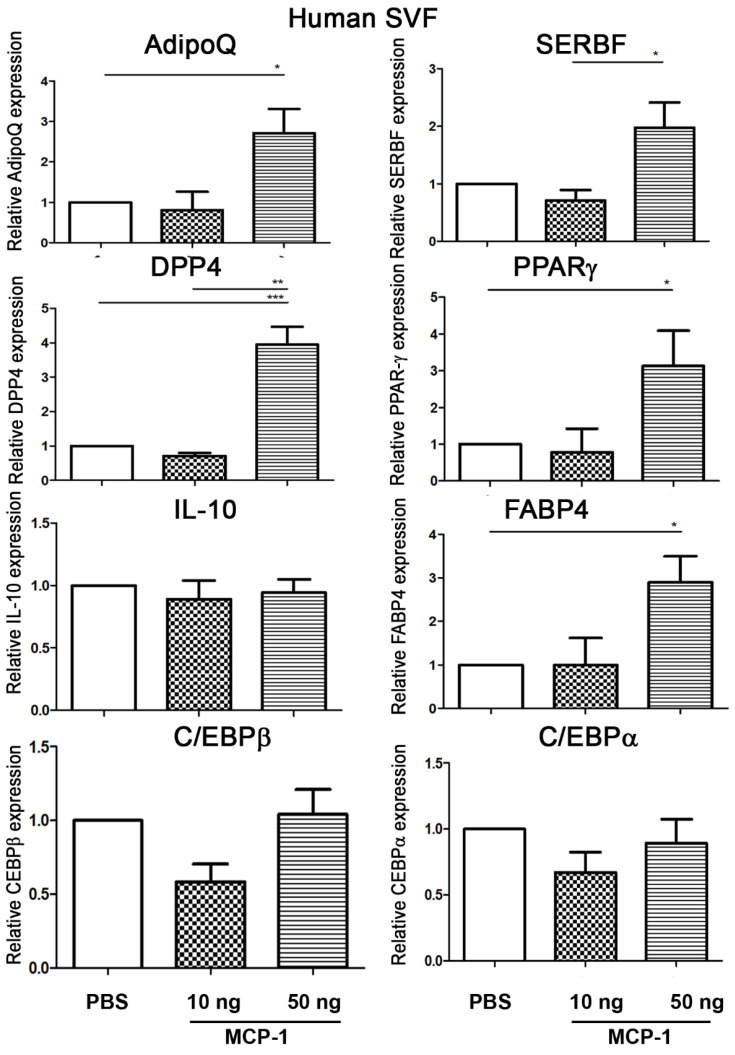
**MCP-1 treatment increases adipogenesis-related gene expression of human adipose SVFs.** We determined the effects of MCP-1 treatment on mRNA expression of adipogenesis-related genes of SVFs from human adipose tissue. Vascular adipose tissue was isolated from subcutaneous human adipose tissue and minced into small pieces. These tissues were then digested with collagenase 8 (Sigma-Aldrich, Cat# C2139) in ice-cold HBSS (2 mg/mL). After passing through a 100 μm cell strainer, cells were centrifuged and the cell pellets were retrieved as SVFs. For in vitro treatment, 1 mL of PBS, or 10 or 50 ng of MCP-1, was added to an equal amount of SVFs (2 × 10^7^ cells), and the mixture was incubated at 37 °C for 3 h. Treatment of human SVFs with 50 ng of MCP-1 resulted in significant increases in AdipoQ, DPP4, PPARγ, FABP4, and SERBF mRNA expression of SVFs compared with the levels of those treated with PBS. N = 5/group. * *p* < 0.05, ** *p* < 0.01, *** *p* < 0.001. FABP4, fatty-acid-binding protein 4; DPP4, dipeptidyl peptidase 4; PPARγ, peroxisome proliferator-activated receptor gamma; PDGFα, platelet-derived growth factor.

**Figure 2 ijms-24-16101-f002:**
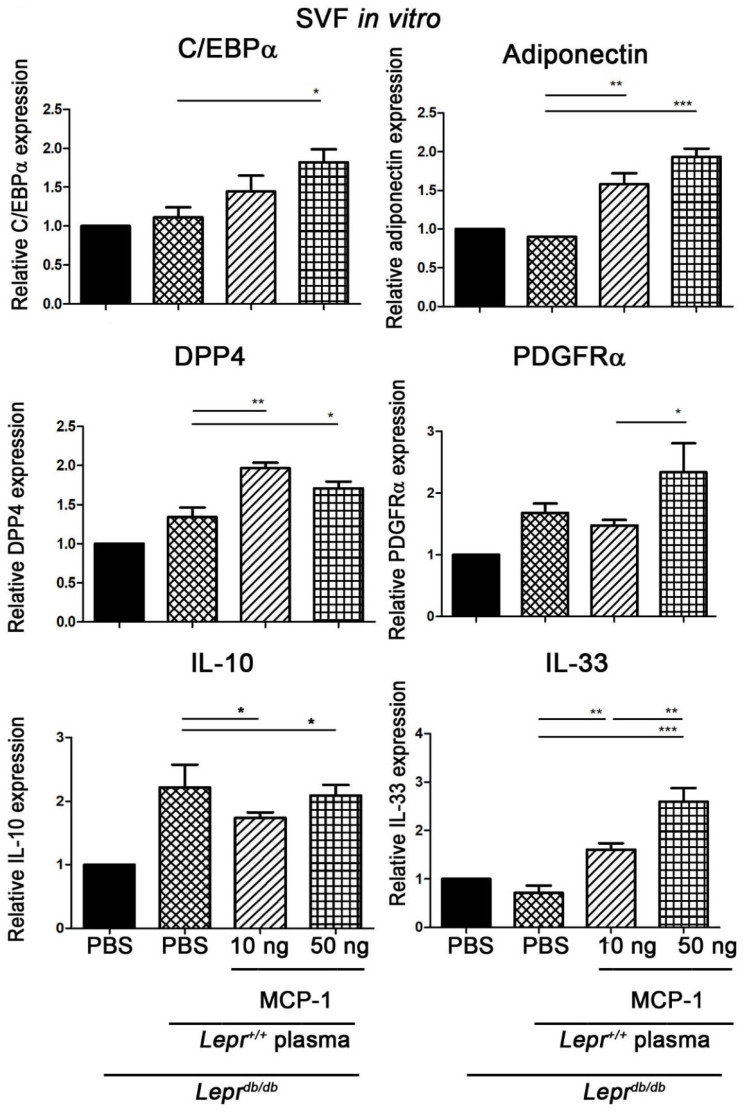
MCP-1-supplemented control plasma increases the expression of adipogenesis-related genes in SVFs from adipose tissue of *Lepr^db/db^* mice. SVFs (2 × 10^7^ cells) were harvested from adipose tissue of *Lepr^db/db^* mice, treated with PBS and 10% *Lepr^+/+^* plasma with or without 10 ng or 50 ng MCP-1 supplementation in vitro, and purified to determine C/EBPα, adiponectin, DPP4, IL-33, PDGFRα, and IL-10 mRNA expression. N = 5/group. * *p* < 0.05, ** *p* < 0.01, *** *p* < 0.001. MCP-1, monocyte chemoattractant protein-1; DPP4, dipeptidyl peptidase 4; C/EBP, CCAAT-Enhancer-binding protein; PDGFR, platelet-derived growth factor receptor.

**Figure 3 ijms-24-16101-f003:**
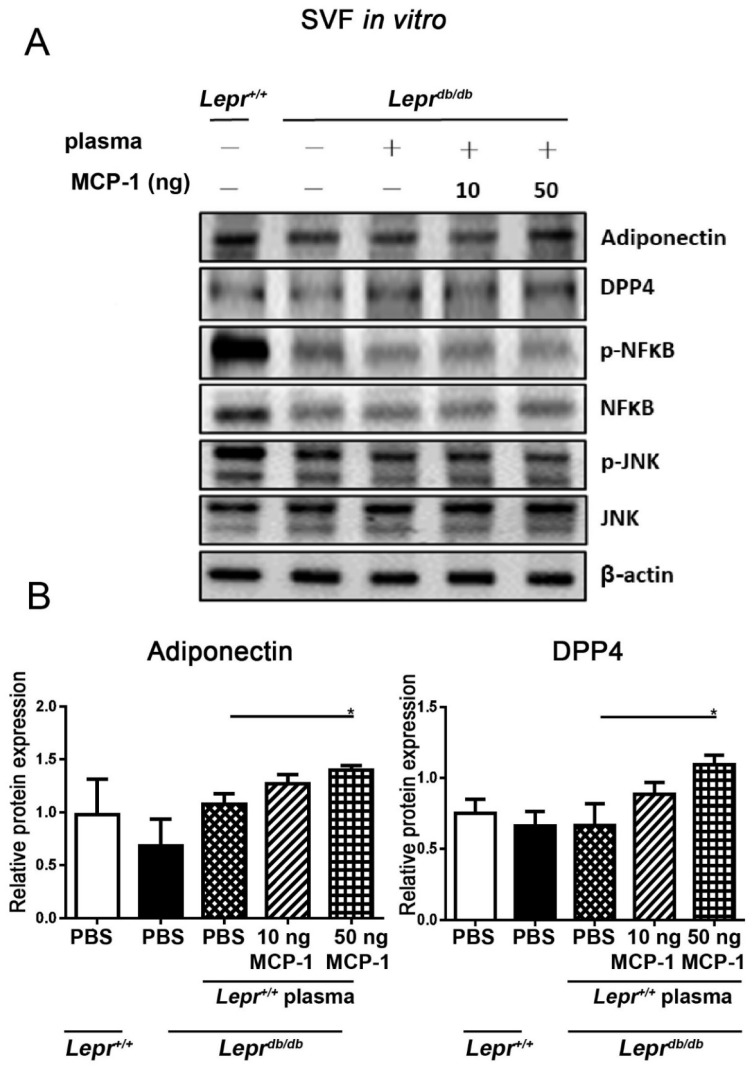
MCP-1-supplemented control plasma increases the protein expression of adiponectin and DPP4 in SVFs from adipose tissue of *Lepr^db/db^* mice. SVFs were harvested from the adipose tissue of *Lepr^db/db^* mice, treated with 10 or 50 ng MCP-1-supplemented *Lepr^+/+^* plasma in vitro, and then purified to determine pNF-κB, NF-κB, pJNK, JNK, adiponectin, and DPP4 protein expression by Western blotting (**A**). The ratios of adiponectin/β-actin and DPP4/β-actin were calculated (**B**). N = 5/group. * *p* < 0.05. MCP-1, monocyte chemoattractant protein-1; C/EBP, CCAAT-Enhancer-binding protein; DPP4, dipeptidyl peptidase 4; PPARγ, peroxisome proliferator-activated receptor gamma; PDGFα, platelet-derived growth factor.

**Figure 4 ijms-24-16101-f004:**
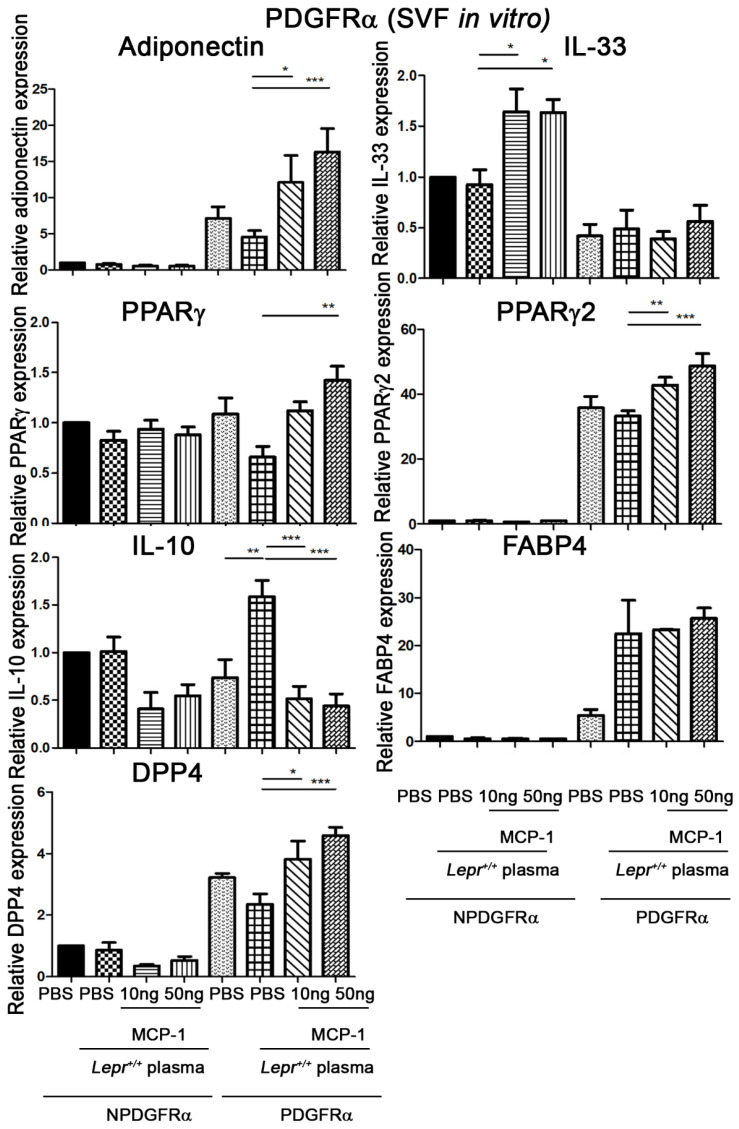
MCP-1-supplemented plasma increases PPARγ, PPARγ2, adiponectin, DPP4, and FABP4 and decreases IL-10 mRNA expression in PDGFRα cells from adipose tissue of *Lepr^db/db^* mice. SVFs (2 × 10^7^ cells) were harvested from adipose tissue of *Lepr^db/db^* mice and treated with PBS and 10% *Lepr^+/+^* plasma with 0, 10, or 50 ng MCP-1 supplementation in vitro. PDGFRα cells were purified with microbeads and analyzed for MCP-1, IL-33, PPARγ, PPARγ2, IL-10, adiponectin, DPP4, and FABP4 mRNA expression by qPCR analysis. N = 5/group. * *p* < 0.05, ** *p* < 0.01, *** *p* < 0.001. NPDGFRα, non-PDGFRα cells; MCP-1, monocyte chemoattractant protein-1; C/EBP, CCAAT-Enhancer-binding protein; DPP4, dipeptidyl peptidase 4; PPARγ, peroxisome proliferator-activated receptor gamma; PDGFα, platelet-derived growth factor; FABP, fatty-acid-binding protein.

**Figure 5 ijms-24-16101-f005:**
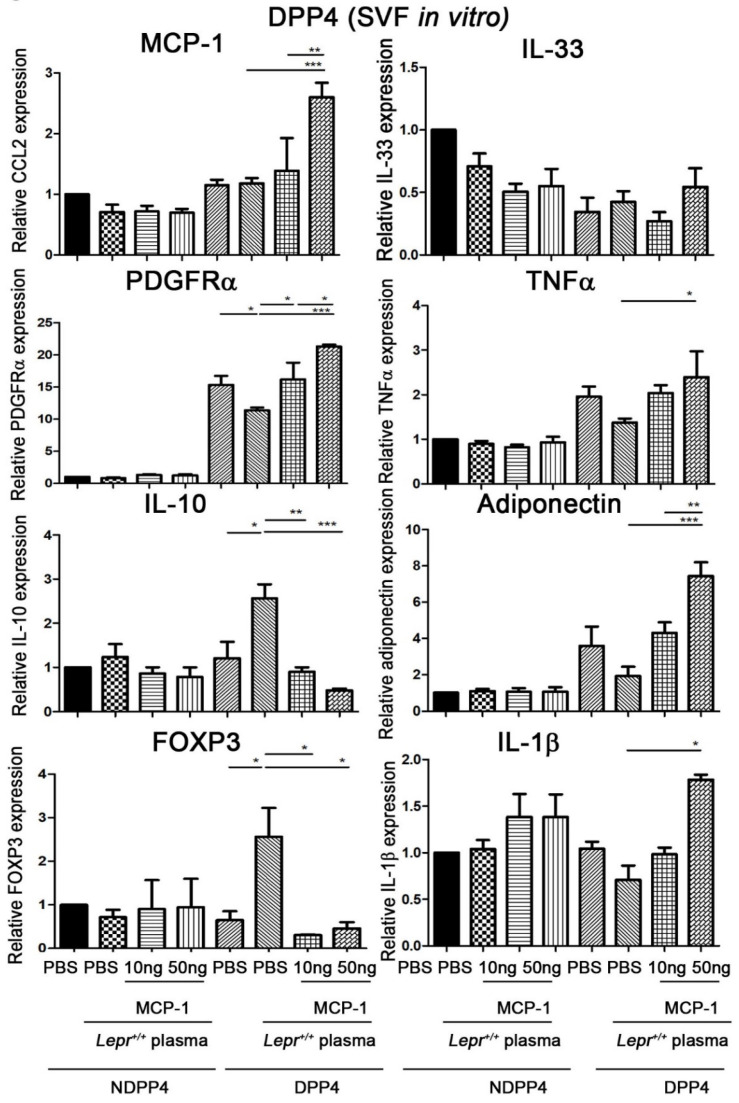
MCP-1-supplemented plasma increases MCP-1, PDGFRα, TNFα, adiponectin, and IL-1β mRNA expression and decreases IL-10 and FOXP3 mRNA expression in DPP4 cells from adipose tissue. SVFs (2 × 10^7^ cells) were harvested from adipose tissue of *Lepr^db/db^* mice and treated with PBS and 10% *Lepr^+/+^* plasma with 0, 10, or 50 ng MCP-1 supplementation in vitro. DPP4 cells were purified with microbeads and analyzed for MCP-1, IL-33, PDGFα, TNFα, IL-10, adiponectin, Foxp3, and IL-1β mRNA expression by qPCR. N = 5/group. * *p* < 0.05, ** *p* < 0.01, *** *p* < 0.001. MCP-1, monocyte chemoattractant protein-1; C/EBP, CCAAT-Enhancer-binding protein; DPP4, dipeptidyl peptidase 4; PPARγ, peroxisome proliferator-activated receptor gamma; PDGFα, platelet-derived growth factor; Foxp3, forkhead box p3.

**Figure 6 ijms-24-16101-f006:**
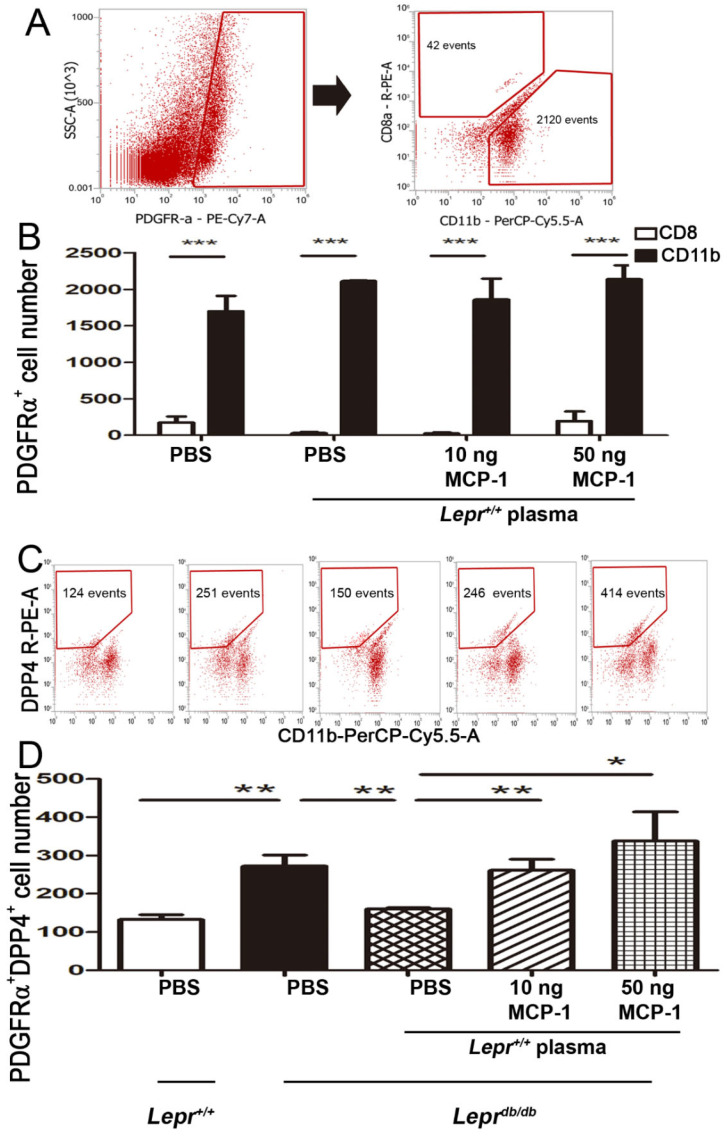
Injection of MCP-1-supplemented plasma into adipose tissue of *Lepr^db/db^* mice increases DPP4^+^ cells among PDGFRα^+^ cells from adipose tissue of *Lepr^db/db^* mice. Plasma was harvested from *Lepr^+/+^* mice, supplemented with 0, 10, or 50 ng of MCP-1, and injected into the adipose tissue of *Lepr^db/db^* mice. The adipose tissue was harvested after 1 week. Flow cytometry was used to assess the numbers of CD8 and CD11b cells in the SVFs from adipose tissue. Cells in the SVFs were suspended in staining buffer (PBS containing 0.5% BSA and 2 mM ethylenediaminetetraacetic acid) and then incubated with CD11b, PDGFRα, and CD8a (BioLegend, clone: 53-6.7) to identify CD11b and CD8a cells among PDGFRα^+^ cells in SVFs (**A**,**B**). In another experiment, cells in the SVFs were incubated with CD11b (BioLegend, clone: M1/70), PDGFRα (BioLegend, clone: APA5), and DPP4 (BioLegend, clone: H194-112) antibodies to identify DPP4 of PDGFRα^+^ cells in SVFs (**C**,**D**). N = 5/group. * *p* < 0.05, ** *p* < 0.01, *** *p* < 0.001. MCP-1, monocyte chemoattractant protein-1; C/EBP, CCAAT-Enhancer-binding protein; DPP4, dipeptidyl peptidase 4; PPARγ, peroxisome proliferator-activated receptor gamma; PDGFα, platelet-derived growth factor.

**Figure 7 ijms-24-16101-f007:**
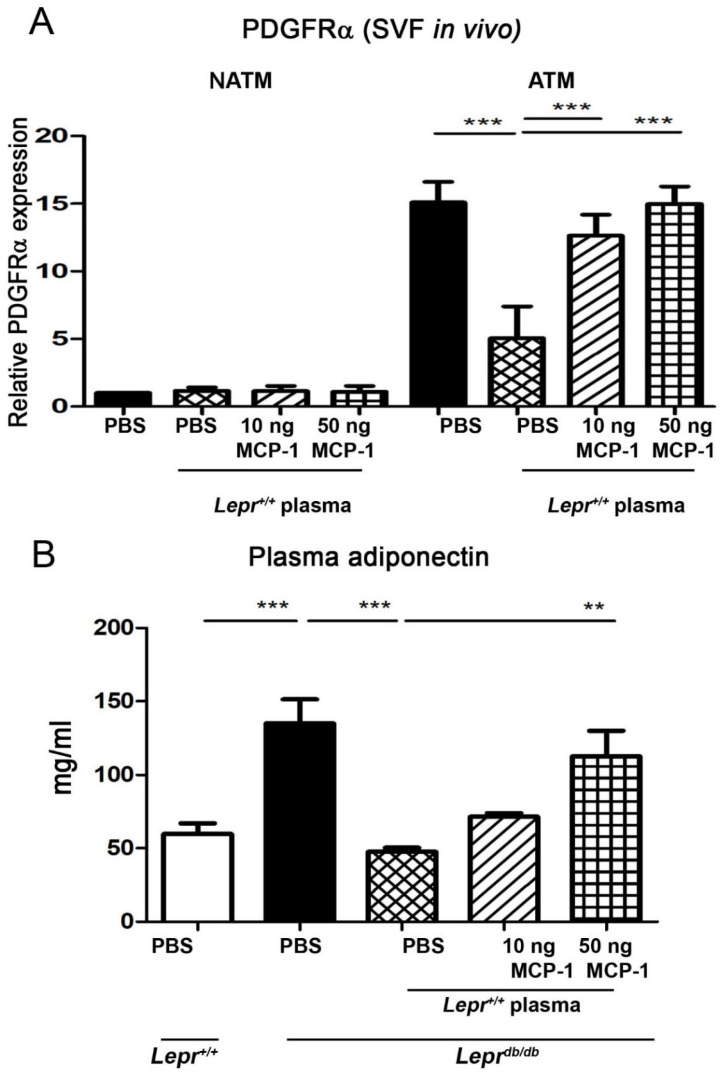
Injection of MCP-1-supplemented plasma into adipose tissue of *Lepr^db/db^* mice increases PDGFRα mRNA expression in adipose tissue macrophages and plasma adiponectin levels. Plasma was harvested from *Lepr^+/+^* mice, supplemented with 0, 10, or 50 ng of MCP-1, and injected into the adipose tissue of *Lepr^db/db^* mice. Adipose tissue macrophages (ATMs) and non-adipose tissue macrophages were purified with microbeads and analyzed for PDGFRα mRNA expression by qPCR (**A**). *Lepr^+/+^* mice were injected with 1 mL PBS into the adipose tissue over the bilateral inguinal area. *Lepr^db/db^* mice were divided into four groups: Group I received 1 mL PBS injection; Group II received an injection of 1 mL of 10% plasma from *Lepr^+/+^* mice; Group III received an injection of 1 mL of 10% plasma from *Lepr^+/+^* mice supplemented with 10 ng MCP-1; and Group IV received an injection of 1 mL of 10% plasma from *Lepr^+/+^* mice supplemented with 50 ng MCP-1. Blood was harvested after 1 week and plasma adiponectin levels were examined by enzyme-linked immunosorbent assay (**B**). N = 5/group. ** *p* < 0.01, *** *p* < 0.01. MCP-1, monocyte chemoattractant protein-1; C/EBP, CCAAT-Enhancer-binding protein; DPP4, dipeptidyl peptidase 4; PPARγ, peroxisome proliferator-activated receptor gamma; PDGFα, platelet-derived growth factor.

## Data Availability

All relevant data and material to reproduce the findings are available in the manuscript. All authors confirm that neither the manuscript nor any parts of its content are currently under consideration or published in another journal.

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
