# Peer review of "Dipeptidyl Peptidase 4 Stimulation Induces Adipogenesis-Related Gene Expression of Adipose Stromal Cells"

_ijms, 2023, doi:10.3390/ijms242216101_

Round 1

Reviewer 1 Report

Comments and Suggestions for Authors

The authors investigated the effects of DPP4 stimulation with monocyte chemoattractant protein-1 (MCP-1) on PDGFRα expression in stromal vascular fraction of adipose tissue and blood adiponectin levels. Treatment of SVFs from human subcutaneous adipose tissues with 50 ng of MCP-1 significantly increased AdipoQ, DPP4, PPARγ, FABP4, and SERBF mRNA expression. MCP-1-supplemented plasma increased adiponectin, C/EBPα, DPP4, IL-33, and PDGFRα mRNA expression and adiponectin and DPP4 protein expression, while decreasing the expression of IL-10 mRNA in SVFs compared with the levels in the plasma treatment group. MCP-1-supplemented plasma was shown to increase PPARγ, PPARγ2, adiponectin, DPP4, and FABP4 and decrease IL-10 mRNA expression in PDGFRα cells from adipose tissue. MCP-1-supplemented plasma increased MCP-1, PDGFRα, TNFα, adiponectin, and IL-1ß and decreased IL-10 and FOXP3 mRNA expression in DPP4 cells. Moreover, the injection of MCP-1-supplemented plasma into adipose tissue increased the proportion of DPP4+ cells among PDGFRα+ cells from adipose tissue and plasma adiponectin levels of Leprdb/db mice. The results demonstrate that DPP4+ cells are important adipose progenitor cells. The authors concluded that DPP4 stimulation with MCP-1 increases adipogenesis-related gene expression and DPP4+ cells among PDGFRα+ cells in SVFs and blood adiponectin levels. Therefore, DPP4 stimulation could be a novel therapy to increase local adipogenesis and systemic adiponectin levels.

Comments

1.       DPP4 inhibitors are widely used in the therapy of type 2 diabetes mellitus. These agents have a neutral effect on bodyweight, which may raise questions about the importance of DPP4 in adipogenesis. This potential contradiction between clinical observations and the results of the study should be discussed.

2.       The authors concluded in the abstract that DPP4 stimulation could be a novel therapy to increase local adipogenesis and systemic adiponectin levels. However, this topic is only briefly mentioned in the Discussion. Is it only a theoretical option, or preliminary result are available?

3.       Legend of Figure 8 is missing.

4.       Description of obtaining human adipose tissue should be specified. These subjects were operated because of lipoma, but the adipose tissue used in the study supposedly not form the lipoma. Were these patients obese? From which area was the tissue obtained?

5.       The list of abbreviations is incomplete.

6.       The number of references seems to be low. Recent publications form the last two years were not cited.

Comments on the Quality of English Language

Minor editing is required. 

Author Response

  1. DPP4 inhibitors are widely used in the therapy of type 2 diabetes mellitus. These agents have a neutral effect on bodyweight, which may raise questions about the importance of DPP4 in adipogenesis. This potential contradiction between clinical observations and the results of the study should be discussed.

Ans: We have added the following in the discussion section: DPP-4 inhibitors are mainly weight-neutral in patients with type 2 diabetes in combination and monotherapy clinical trials. This effect may contribute to the inhibition of intestinal fat extraction [1]. Out data demonstrate that DPP4 stimulation with MCP-1 increases adipogenesis-related gene expression and DPP4+ cells among PDGFRα+ cells in SVFs and blood adiponectin levels. This suggests that DPP-4 stimulation may induce local adipogenesis through the increase of adipogenesis gene expression without weight gain.

  1. The authors concluded in the abstract that DPP4 stimulation could be a novel therapy to increase local adipogenesis and systemic adiponectin levels. However, this topic is only briefly mentioned in the Discussion. Is it only a theoretical option, or preliminary result are available?

Ans: We have added the following in the discussion section: Radiation-induced soft tissue defects have become a common problem in cancer treatment[2]. However, there were only limited methods could be used to increase local adipogenesis or fat grafting survival. The PPARγ agonist thiazolidinedione (TZD) is used as a proadipogenic compound, but its use remains controversial because it is associated with weight gain and certain cardiac side effects [3]. Adiponectin treatment reverses high-fat-diet-induced insulin resistance through the increase of insulin-stimulated glucose in adipose tissue and muscles in mice [3]; however, adiponectin agonists have not been well characterized and their availability for therapeutic purpose in humans is still limited. Taken together, our findings suggest that MCP-1-supplemented plasma stimulates adipogenesis-related gene expression and adiponectin levels through increases of DPP4 and PDGFRα. MCP-1-supplemented plasma may be used as a new therapeutic strategy to increase local adipogenesis in treating lipodystrophy and soft tissue defects.

  1. Legend of Figure 8 is missing.

Ans: We have deleted the Figure 8.

  1. Description of obtaining human adipose tissue should be specified. These subjects were operated because of lipoma, but the adipose tissue used in the study supposedly not form the lipoma. Were these patients obese? From which area was the tissue obtained?

Ans: We have changed the paragraph related to the human subjects as the followings:  Human adipose tissue was obtained from normal subcutaneous adipose tissue during an operation involving excision of lipoma from six female patients (aged 20-40 years, BMI 19.0-25.0 kg/m2),

  1. The list of abbreviations is incomplete.

Ans: The list of abbreviations has been expanded.

  1. The number of references seems to be low. Recent publications form the last two years were not cited.

Answer: We have added a few recent publications to the references.

Reviewer 2 Report

Comments and Suggestions for Authors

The authors investigated the effects of DPP4 with MCP-1 on the expression of the adipogenesis-related genes in SVFs. MCP-1 enhanced the expression of AdipoQ, DPP-4, PPARγ, FABP4, and SERBF. Moreover, MCP-1 administration elevated the mRNA levels of AdipoQ, C/EBPα, DPP-4, IL-33, and PDGFRα, while lowered the expression of IL-10. In addition, MCP-1 increased the expression of PPARγ, PPARγ2, AdipoQ, DPP4, and FABP4, but decreased IL-10 expression in PDGFRα cells of adipose tissues. While, MCP-1 enhanced the expression of MCP-1, PDGFRα, TNFα, AdipoQ, and IL-1β, but decreased IL-10 and FOXP3 mRNA expression in DPP4 cells. Th results are basically sound. However, there are concerns that should be addressed

1.         The abbreviated words should be shown by the full-spelling, when they were first appeared.

2.         When the SVFs were treated with MCP-1, is there anything changes in phenotype? The pictures of the cells had better be shown, if phenotypes in the cells are changed.

3.         In Fig. 3, NF-κB consists of subfamily, including p50, p65, and c-Rel etc. The authors should show which NF-κB was investigated.

4.         In the legend of Fig. 3, **p<0.01 is not found in the figure.

5.         Possible mechanism that MCP-1 induces the expression of the adipogenesis-related genes in SVFs should be stated in Discussion part.

6.         DPP-4 expression is increased by MCP-1 in SVFs. What mechanism is involved in this regulation? If possible, please state in Discussion part.

7.         MCP-1-induced expression of the adipogenesis-related genes is mediated by increasing in the DPP-4 expression. If DPP-4 expression is suppressed by its siRNA or CRISPR method, the enhancement of these gene expressions is cancelled? It had better be confirmed.

8.         It is known that MCP-1 is an inflammation-related protein that induces monocyte infiltration into inflammation sites, and its level is increased in obese adipose tissues. However, in obese adipose tissue, adiponectin level is decreased. This discrepancy should be discussed.

Author Response

  1. The abbreviated words should be shown by the full-spelling, when they were first appeared.

Ans: The abbreviated words have been shown by the full-spelling when they were first appeared.

  1. When the SVFs were treated with MCP-1, is there anything changes in phenotype? The pictures of the cells had better be shown, if phenotypes in the cells are changed.

Ans: We found that there were no significant differences of CD11b cells and CD8 cells in SVFs between control and different treatment groups (Fig. 6A and B).

  1. In Fig. 3, NF-κB consists of subfamily, including p50, p65, and c-Rel etc. The authors should show which NF-κB was investigated.

Ans: pNF-κB P65 (Cell signaling, # 3033) has been used in Western blotting. This has been revised in method section.  

  1. In the legend of Fig. 3, **p<0.01 is not found in the figure.

Ans: In the legend of Fig. 3, **p<0.01 has been deleted.

  1. Possible mechanism that MCP-1 induces the expression of the adipogenesis-related genes in SVFs should be stated in Discussion part.

Ans: Thanks for reviewer’s suggestion. We have added the following in the discussion section: Macrophages in the visceral adipose tissue demonstrated increased proliferation in obesity. MCP-1 treatment induced macrophage cell division in adipose tissue, whereas MCP-1 deficiency in vivo reduced adipose tissue proliferation [4]. This may be the reason why MCP-1 induces the expression of the adipogenesis-related genes in SVFs.

  1. DPP-4 expression is increased by MCP-1 in SVFs. What mechanism is involved in this regulation? If possible, please state in Discussion part.

Ans: We have added the following in the discussion section: Macrophages within the visceral adipose tissue displayed increased proliferation in obesity. MCP-1 treatment induced macrophage cell division in adipose tissue explants, whereas mcp-1 deficiency in vivo decreased adipose tissue proliferation [4]. DPP4-positive interstitial progenitor cells contribute to basal and high-fat-diet-induced adipogenesis [5]. Therefore, we suggest that MCP-1 induced DPP4 expression in stromal vascular cells.

  1. MCP-1-induced expression of the adipogenesis-related genes is mediated by increasing in the DPP-4 expression. If DPP-4 expression is suppressed by its siRNA or CRISPR method, the enhancement of these gene expressions is cancelled? It had better be confirmed.

Ans: We have added the followings in the discussion section: A targeted reduction of DPP-4 expression in the liver with siRNAs reduced hepatic mRNA expression of PPARγ. This suggests that a targeted reduction of DPP-4 expression in the liver may improve hepatic lipid metabolism [6]. Mice lacking the DPP-4 gene were difficult to develop obesity after a high-fat diet [7]. Ablation of the DPP-4 gene was also associated with reduced lipogenesis [7].

  1. It is known that MCP-1 is an inflammation-related protein that induces monocyte infiltration into inflammation sites, and its level is increased in obese adipose tissues. However, in obese adipose tissue, adiponectin level is decreased. This discrepancy should be discussed.

Ans: Thanks for reviewer’s suggestion. We have added the followings in the discussion section: MCP-1 induced adipogenesis related gene expression of stromal vascular cells. Defects in adipocyte differentiation drive pathologic adipose tissue remodeling, fibrosis, immune cell activity, and metabolic syndrome. Adipogenesis supports healthy adipose tissue remodeling in obesity. Stimulating adipogenesis can drive healthy WAT remodeling. Therefore, healthy WAT remodeling contributes to the maintenance of adiponectin levels in obesity [8].

  1. Foley JE, Jordan J. Weight neutrality with the DPP-4 inhibitor, vildagliptin: mechanistic basis and clinical experience. Vasc Health Risk Manag. 2010;6:541-8. doi: 10.2147/vhrm.s10952. PubMed PMID: 20730070; PubMed Central PMCID: PMCPMC2922315.
  2. Kenny EM, Egro FM, Ejaz A, Coleman SR, Greenberger JS, Rubin JP. Fat Grafting in Radiation-Induced Soft-Tissue Injury: A Narrative Review of the Clinical Evidence and Implications for Future Studies. Plast Reconstr Surg. 2021;147(4):819-38. doi: 10.1097/PRS.0000000000007705. PubMed PMID: 33776031.
  3. Li X, Zhang D, Vatner DF, Goedeke L, Hirabara SM, Zhang Y, et al. Mechanisms by which adiponectin reverses high fat diet-induced insulin resistance in mice. Proc Natl Acad Sci U S A. 2020;117(51):32584-93. Epub 2020/12/10. doi: 10.1073/pnas.1922169117. PubMed PMID: 33293421; PubMed Central PMCID: PMCPMC7768680.
  4. Amano SU, Cohen JL, Vangala P, Tencerova M, Nicoloro SM, Yawe JC, et al. Local proliferation of macrophages contributes to obesity-associated adipose tissue inflammation. Cell metabolism. 2014;19(1):162-71. doi: 10.1016/j.cmet.2013.11.017. PubMed PMID: 24374218; PubMed Central PMCID: PMCPMC3931314.
  5. Stefkovich M, Traynor S, Cheng L, Merrick D, Seale P. Dpp4+ interstitial progenitor cells contribute to basal and high fat diet-induced adipogenesis. Mol Metab. 2021;54:101357. doi: 10.1016/j.molmet.2021.101357. PubMed PMID: 34662714; PubMed Central PMCID: PMCPMC8581370.
  6. Gorgens SW, Jahn-Hofmann K, Bangari D, Cummings S, Metz-Weidmann C, Schwahn U, et al. A siRNA mediated hepatic dpp4 knockdown affects lipid, but not glucose metabolism in diabetic mice. PLoS One. 2019;14(12):e0225835. doi: 10.1371/journal.pone.0225835. PubMed PMID: 31794591; PubMed Central PMCID: PMCPMC6890245 shareholders of Sanofi. This does not alter the authors' adherence to PLOS ONE policies on sharing data and materials.
  7. Conarello SL, Li Z, Ronan J, Roy RS, Zhu L, Jiang G, et al. Mice lacking dipeptidyl peptidase IV are protected against obesity and insulin resistance. Proc Natl Acad Sci U S A. 2003;100(11):6825-30. doi: 10.1073/pnas.0631828100. PubMed PMID: 12748388; PubMed Central PMCID: PMCPMC164531.
  8. Vishvanath L, Gupta RK. Contribution of adipogenesis to healthy adipose tissue expansion in obesity. J Clin Invest. 2019;129(10):4022-31. Epub 2019/10/02. doi: 10.1172/JCI129191. PubMed PMID: 31573549; PubMed Central PMCID: PMCPMC6763245.

Round 2

Reviewer 2 Report

Comments and Suggestions for Authors

The manuscript was improved. 

Author Response

ijms-2643103 Minor revision

Dipeptidyl peptidase 4 stimulation induces adipogenesis-related gene expression in adipose stromal cells

Dear Sir:

We have

(I) checked that all references are relevant to the contents of the manuscript.

(II)  Highlighted the revisions of the manuscript.

(III) Also, we have point by point revised the manuscript according to reviewers’ suggestion.

Thanks for your help

Best regards

.

Dr. Lee-Wei Chen

Department of Surgery, Kaohsiung Veterans General Hospital, 386, Ta-chung 1st Road, Kaohsiung, Taiwan

Fax: 8867-3455064

Phone: 8867-3422121 ext 3029

E-mail address: lwchen@vghks.gov.tw